# An Efficient Combination of Convolutional Neural Network and LightGBM Algorithm for Lung Cancer Histopathology Classification

**DOI:** 10.3390/diagnostics13152469

**Published:** 2023-07-25

**Authors:** Esraa A.-R. Hamed, Mohammed A.-M. Salem, Nagwa L. Badr, Mohamed F. Tolba

**Affiliations:** 1Faculty of Computer and Information Sciences, Ain Shams University, Cairo 11566, Egypt; nagwabadr@cis.asu.edu.eg (N.L.B.); fahmytolba@cis.asu.edu.eg (M.F.T.); 2Media Engineering and Technology, German University in Cairo (GUC), Cairo 16482, Egypt; mohammed.salem@guc.edu.eg

**Keywords:** lung cancer, histopathological, squamous cell carcinomas, light gradient boosting, LightGBM

## Abstract

The most dangerous disease in recent decades is lung cancer. The most accurate method of cancer diagnosis, according to research, is through the use of histopathological images that are acquired by a biopsy. Deep learning techniques have achieved success in bioinformatics, particularly medical imaging. In this paper, we present an innovative method for rapidly identifying and classifying histopathology images of lung tissues by combining a newly proposed Convolutional Neural Networks (CNN) model with a few total parameters and the enhanced Light Gradient Boosting Model (LightGBM) classifier. After the images have been pre-processed in this study, the proposed CNN technique is provided for feature extraction. Then, the LightGBM model with multiple threads has been used for lung tissue classification. The simulation result, applied to the LC25000 dataset, demonstrated that the novel technique successfully classifies lung tissue with 99.6% accuracy and sensitivity. Furthermore, the proposed CNN model has achieved the lowest training parameters of only one million parameters, and it has also achieved the shortest processing time of just one second throughout the feature extraction process. When this result is compared with the most recent state-of-the-art approaches, the suggested approach has increased effectiveness in the areas of both disease classification accuracy and processing time.

## 1. Introduction

In the modern world, cancer is among the most terrible diseases that adversely damage and endanger a person’s life. In 2020, according to the World Health Organization, cancer is predicted to be the top cause of death globally [1]. Lung cancer accounts for an estimated 1.80 million cancer-related deaths globally. According to projections, the number of cancer-related deaths could reach 60% by 2035 [2]. Lung tumors are masses of lung cells that have transformed and developed out of control. Lung cancer incidence has increased across the globe for a variety of reasons. The reasons include exposure to toxic or hazardous substances inhaled, and a high percentage of older individuals in society have suffered from this. Smoking cigarettes is thought to account for 70–80% of lung cancer risks in women and 90% of lung cancer risks in men [3]. Although people who have never smoked can develop lung cancer, the risk is generally higher. Lung cancers have adenocarcinomas and squamous cell carcinomas as their most common subtypes. Small and large cell carcinomas are other histologic subtypes. However, both large and small cell carcinomas tend to spread fast and can develop in any section of the lung, making therapy more challenging [4]. Squamous cell carcinoma develops when the uncontrolled growth of aberrant lung cells results in a tumor. One organ where cancer cells may spread (metastasize) is the lymph nodes in and around the lungs, liver, adrenal glands, bones, and brain. Normally, squamous cell carcinoma develops in the central lungs, and unless it is identified and managed extremely promptly, it commonly develops all across the body [5].

Lung cancer, consequently, has among the highest incidence and death rates from any major cancer worldwide [1]. The fight against lung cancer depends on the early diagnosis of worrisome lung nodules. To determine the type of cancer, one of the most important aspects is the histopathological diagnosis. Analysis of lung cancer histopathology images is urgently required, since how the cancer is treated depends on the stage of the disease, the molecular profile, and the type of tissue [6].

Recently, autonomous cancer diagnosis using machine learning and deep learning approaches has advanced tremendously. They are, therefore, used to reduce the workload for pathologists and hasten the crucial process of discovering lung cancer. Its fundamental objective is to make it possible for computers to recognize, classify, and analyze visual information in a manner comparable to that of humans, and then to use that data to generate the necessary results. Because it allows for quicker diagnosis as well as improved treatment response, cancer detection in its early stages increases overall survival for many individuals and may even save lives [7].

One deep learning technique that may be used to identify an image is CNN [8]. As a result of its efficiency in recognizing feature representations, CNN is extensively utilized in image processing. With high accuracy, deep learning has been used in several biological domains. This achievement’s foundation, CNN, is built on the multi-layer feature extraction from the data itself [9,10]. A generation network that reconstructs medical images from the segmentation network’s predictions is offered by generative consistency [11] as an alternative to directly promoting consistency on network segmentation outcomes. Weak annotation also comprises image-level annotation, sparse annotation, and noisy annotation in addition to partial annotation [12]. The creation of self-adapting frameworks has promise as well, as demonstrated by the nnU-Net [13], which has demonstrated outstanding performance in several medical image segmentation tasks. By choosing the best option for a number of steps, including preprocessing, hyperparameter optimization, architecture, etc., this framework adapts specifically to the task at hand. It is likely that a similar optimization framework would perform well for classification or localization tasks, including those for CXR images. The classification of numerous diseases, including brain disorders [14], the detection of breast cancer [15], skin cancer [16], arrhythmia detection [17], the detection of pulmonary pneumonia in X-ray images [18], the segmentation of fundus images [19], and the segmentation of the lung [20], has advanced significantly due to CNN.

The primary contributions made by this study are the pre-processing LC25000 approaches used to improve the contrast between the images after they have been extracted from the lung histopathology image dataset. The efficient proposed CNN model with the lowest training parameter for obtaining discriminative feature vectors is then used to perform the feature extraction. The LightGBM with the fastest computation time was utilized to classify lung tissue to increase the efficiency and accuracy of disease classification. This study’s innovation is a combination of the proposed CNN-LightGBM strategy that can fast recognize and categorize histological images of lung tissues. It also minimizes the false negative rate, which lowers the risk of incorrectly concluding that the disease of a patient does not exist. We then compare the proposed model with existing machine learning and deep learning techniques. The proposed CNN-LightGBM strategy is more effective than cutting-edge methods for feature extraction and classification of histopathological lung cancer datasets.

The rest of this paper is organized as follows: Section 2 introduces the literature review of lung cancer classification, followed by the details of the histopathology lung cancer datasets that are used. The data pre-processing steps applied to the selected dataset to make it ready for training by the proposed CNN model are then outlined in Section 3. Then, in Section 4, we discuss the proposed CNN-LightGBM for lung cancer histopathology detection and classification. We go over the findings of our experiments in Section 5. Subsequently, the suggested CNN-LightGBM model’s effectiveness in feature extraction and classification is compared to a few existing deep learning and machine learning models. In Section 6, the summary of the work as well as recommended future work are presented.

## 2. Related Work

The use of machine learning (ML) and deep learning (DL) for categorization and identification purposes has been a hot topic for a while. In recent years, artificial intelligence (AI) technology based on CNN has been widely applied in various fields [21,22]. This study evaluates the proposed method using the LC25000 histopathology imaging dataset of lung cancer, which was released in 2020. This section presents the work of several researchers that have used this dataset to create deep learning-based applications.

Using the LC25000 dataset, authors in [6] automated the detection of colon and lung cancer. The pre-processing of the channel-separated images included wavelet decomposition and the 2D Fourier transform. They achieved a 96.37% accuracy using a CNN model. Authors in [23] classified LC25000 lung cancer histology images using CNN. ResNet50, Inception ResNet V2, DenseNet121, and VGG19 were used to extract features. Three hidden layers of CNN were employed to categorize the images. With a 99.7% accuracy rate, Inception-ResNetv2 performed significantly better.

Homology-based image processing techniques were suggested by the authors in [24]. They proposed looking at conventional texture-based image processing techniques as well. Binarization, grayscale transformation, and Betti number conversion were used to modify the appearance of the images. The accuracy was 99.43%. Adenocarcinomas of the colon and the lungs were classified by [25] using a CNN model. For this study, they made use of the LC25000 dataset. The images were first reduced in size to 150 × 150 pixels, after which they underwent randomized shear and zoom modifications before being normalized. A CNN model was applied separately for the lung dataset and the colon dataset, producing accuracy readings of 97% and 96%, respectively.

The authors in [26] proposed a CNN model on the LC25000 dataset to classify malignant from normal cells in the colon, using Lime and Deep Lift as optimization approaches, and more than 94% accuracy was attained. The LC25000 dataset for colon cancer was analyzed by authors in [27] using MobileNetV2 and CNN models with max-pooling and average pooling layers. The accuracy of the MobileNetV2 and CNN models with maximum and average pooling was 97.49%, 95.48%, and 99.67%, respectively.

The authors in [28] developed four different CNN models for the classification of lung cancer. The input images were considered in three distinct sizes. The highest accuracy on the test dataset was 96.6%, using an input size of 768 × 768 pixels and a CNN model with four convolutional layers and maximum pooling layers. It was found that accuracy increased as input image size and convolutional layer count increased.

The authors in [29] used various feature extraction methods. For example, they used VGG16, InceptionV3, ResNet50, etc. to develop eight pre-trained CNN models for lung and colon cancer classification. They achieved accuracy levels between 96% and 100%. For the classification of images of lung cancer, the authors in [30] constructed a CNN model with cross-entropy as an error function. They attained 97.2% validation accuracy and 96.11% training accuracy. The authors in [31] employed a CNN model with gamma correction, with gamma values of 0.8, 1.0, and 1.2. With a gamma value of 1.2, maximum accuracy of 87.16% was achieved. A unique approach was constructed to automatically classify the LC25000 lung histology image collection [32]. The accuracy rating for the EGOA (Enhanced Grasshopper Optimization Algorithm) with random forest model was 98.50%. EffcientNetV2 big, medium, and small models are a deep learning architecture built on the concepts of compound scaling and progressive learning [33]. Using the EffcientNetV2-L model for the 5-class categorization of lung and colon cancers, they attained an accuracy of 99.97% on the test set.

Hyperparameters are the factors that affect how the network is trained as well as the variables that govern the network’s topology. Before training (and before maximising the weights and bias), hyperparameters are established. The suggested model, which only uses four convolutional layers, four maximum pooling layers, and one leaky layer, gives us the greatest accuracy with the fewest total parameters out of all the models we tried to build. It can be argued that the proposed CNN-LightGBM with the multiple threads model performs better than the majority of the current models. With the quickest computation time of only three seconds and the fewest number of parameters needed to identify and classify lung tissue, it was able to reach an accuracy of up to 99.6%.

## 3. The Datasets

The histopathology imaging dataset for lung and colon cancer, LC25000, was published in 2020. It was used to analyze the proposed technique in this work. There are a significant number of lung and colon cancer diagnoses, and the LC25000 dataset is the recent generation, which has an adequate number of images for deep learning. Therefore, numerous investigators have just implemented deep learning-based applications in his dataset. The LC25000 Lung and Colon Histopathological Imaging Collection are divided into five categories, as shown in Table 1. The information has been verified and complies with HIPAA [34].

The total number of original images gathered is only 750 for lung tissue and 500 for colon tissue. It also has 250 images in each category with a resolution of 1024 × 768 pixels. These images are scaled down to 768 × 768 pixels using Python and then expanded using the augmenter software package.

As shown in Table 1, the lung and colon datasets have 5000 images in each category. The left and right rotations (up to 25 degrees, 1.0 probability), together with the horizontal and vertical flips, are all used for augmentation [34]. Figure 1 displays the sample images for each category.

### 3.1. The Selected Dataset

A total of 15,000 digitized images of lung cancer histopathology slides were made accessible from the LC25000 databases. Squamous cell carcinomas (SCC) of the lung are categorized as non-small cell lung cancers (NSCLC). The major airways, such as the left or right bronchus or the center of the lung, are common sites for the development of squamous cell lung cancer. Cellular transformation is often caused by cigarette smoke. Almost 70 to 80% of instances of lung cancer in women and nearly 90% of cases in males are attributed to smoking. SCC is more strongly associated with smoking than other NSCLC subtypes. Other risk factors for SCC include age, family history, mineral and metal particle exposure, asbestos exposure, and secondhand smoking.

Therefore, we suggested separating images of benign lung tissue from those of SCC of the lung. The authors in [7] have demonstrated that the selected dataset is balanced. It has been tested in four experiments, each with four randomly selected samples of data, and the resulting standard deviation error was quite low.

### 3.2. Data Pre-Processing

Using the Augmenter tool [34], the LC25000 histopathology datasets have been augmented. The data that a neural network uses to train on has a significant impact on how accurate an image classification model is. By eliminating all of the noise and disruption in the input image, image pre-processing allows us to focus on the characteristics that the neural network should learn. Image recognition techniques are contingent upon the quality of the dataset.

The pre-processing procedures have been carried out by enlarging the image to 256 × 256, converting it to bgr2rgb, and then converting it to a NumPy array, as illustrated in Figure 2. The following step is feature scaling, which involves applying the generalization approach to the image. After that, performing labeling has been performed by giving each image a label (lung_n or lung_scc). Then, the generalization approach is applied to the image to perform feature scaling, where the values of an easy image array have been divided by 255, which is the highest possible value (the maximum intensity value of an image).

## 4. The Proposed Approach

The proposed CNN-LightGBM with the multiple threads model applied to LC25000 histopathology lung cancer image classification is presented in depth in this section. The suggested architecture consists of two steps: feature extraction and image classification, as shown in Figure 3. Pre-processing of images is the first stage in this framework’s procedure to guarantee that every image in the dataset has consistency in terms of color, size, and variance. Our suggested solution handled this by transforming the image into bgr2rgb, converting the image to a NumPy array, performing feature scaling, and, finally, assigning a label to the image.

Every step has been made to guarantee the image quality before sending it to the suggested CNN feature extraction model. The suggested CNN model then acts as an essential component in extracting all of the important features from an image to gain more knowledge. The proposed framework’s final step trains a LightGBM multiple threads classifier (four threads) with all of the retrieved features. With this boosting strategy, we can train our suggested network to classify distinct types of lung histology images in a scalable and highly effective manner. Finally, testing data has been classified into benign or malignant lung cancer using the parameters of the trained model.

### 4.1. The Proposed Deep Learning Feature Extractor

The feature extraction strategy has a substantial influence on how well the classification process performs. It is an essential step in understanding the features present in images of lung cancer histology. The DL approach will automatically detect an image’s attributes based on each pixel in it. This is the approach we have chosen to take. Based on the idea of a CNN, convolution performs discrete spatial processing operations that are conveniently calculated as discrete spatial processing processes [35].

Layers employed for image recognition and classification are stacked to create CNN’s architecture. Before going through the fully connected layer, training and testing data are transmitted through filters such as max-pooling and kernel filters. All of the hidden layers made use of the activation function ReLU. Layers and parameters are discussed as follows:

**Input Layer:** With this layer, data were put into and passed into the first convolution layer. In this case, the input is a 256 × 256 pixel image with RGB color channels. **Convolution Layer:** To understand the geographic structure of images, this layer was employed to convolute the input image with trainable filters. This model has four convolution layers (CL), each with kernels of 32, 64, 128, and 256. The first CL layer to be utilized has a kernel size of (11 × 11) and a stride of 4. The padding has also been set to valid. The kernel size for the other CL levels is (3 × 3), with a stride of 1, and the padding has been kept the same. ReLU activation has also been used to boost performance in nonlinear procedures.**Pooling Layer:** The output convolution layer images have been down-sampled using a pooling process. There is a max pooling layer with a (2 × 2) kernel size utilized after each CL layer. The most popular max pooling operation has been employed by all pooling layers.**Flatten Layer:** The output of the convolution layer has been converted using this layer into a 1D tensor.**Fully connected layer or dense layer:** With a simple vector as their input, these layers produce a vector as their output. There are two dense layers in this model. In the first dense layer, there are 1024 neurons. In the second dense, there are 512 neurons. **Dropout Layer:** A dropout layer, which randomly removes neurons from both visible and hidden levels, has been used between fully connected layers to prevent the model layers from getting overfitted. This layer’s rate is 0.4. Table 2 and Figure 4 show the proposed CNN architecture.

### 4.2. LightGBM Model

In 2017, Microsoft released LightGBM, a data model based on gradient-boosting decision trees (GBDT) [36]. GBDT combines weak learners to create a strong one. The decision tree for the GBDT method, however, can only be a regression tree, since each subsequent tree in the process takes the results and residuals of all preceding trees, as shown in Figure 5. A current residual regression tree is produced by using the residual of each projected outcome and desired value as the goal of additional learning. The final anticipated result is the sum of the outcomes from each decision tree [37]. The recent geometric expansion in data volume necessitates the adjustment of accuracy and efficiency, even though GBDT has achieved positive learning results on a variety of machine learning applications. The LightGBM [36] algorithm has recently been suggested. It uses less memory and considerably speeds up forecasting while preserving prediction accuracy. The classic GBDT approach usually requires more calculation time than necessary to create a decision tree.

The best segmentation point must be identified before building a decision tree. The standard approach is to first sort feature values before enumerating every available feature point. This process is time consuming and memory intensive. An enhanced histogram method is used by the LightGBM algorithm. The continuous eigenvalues are divided into k intervals by selecting division points from among the k values. Consequently, in terms of training time and space efficiency, it exceeds the GBDT algorithm. At the same time, the decision tree is a poor classifier. The histogram technique can successfully prevent overfitting by having a regularization impact. The LightGBM method employs a leaf-wise generation approach to reduce training data. While growing the same leaf, the leaf-wise strategy can cut losses more than the more conventional level-wise method that is shown in figure. Moreover, the additional parameter is employed to restrict the decision tree’s depth and prevent overfitting.

## 5. The Experimental Work and Results

The CNN-LightGBM approach has been employed in this study to distinguish between images of benign lung tissue and squamous cell carcinoma (SCC) of the lung. It has been applied to the LC25000 lung histology images dataset. A Python 3 Google Compute Engine backend (GPU), NVIDIA T4 Tensor Core GPUs, with 12.68 GB of RAM and 78.19 GB of disc space, was used to test the model in Google Colab.

### 5.1. Performance Measures

Several metrics have been used to evaluate machine learning models. The metrics include the confusion matrix and related metric parameters. The confusion matrix is frequently used to evaluate classification processes. The actual class and predicted class are arranged in a two-by-two matrix for binary classification. The proportion of predicted observations to all observations is known as accuracy, and it is the basic performance statistic used in medical image classification. The percentage of negative and positive cases is how specificity and sensitivity are expressed. Equations (1)–(5) provide the mathematical formulas of precision, accuracy, F1 score, sensitivity, specificity, and Matthew Correlation Coefficient (MCC), respectively.
(1)$$\mathrm{Accuracy}=\frac{\mathrm{TP}+\mathrm{TN}}{\mathrm{TN}+\mathrm{FP}+\mathrm{FN}+\mathrm{TP}},$$
(2)$$\;F1\text{-}\mathrm{Score}=\frac{2\times \mathrm{TP}}{\left(2\times \mathrm{TP}\right)+\mathrm{FP}+\mathrm{FN}},$$
(3)$$\mathrm{Sensitivity}=\frac{\mathrm{TP}}{\mathrm{TP}+\mathrm{FN}},$$
(4)$$\mathrm{Specificity}=\frac{\mathrm{TN}}{\mathrm{TN}+\mathrm{FP}}$$
(5)$$\mathrm{MCC}=\frac{\mathrm{TP}\times \mathrm{TN}-\mathrm{FP}\times \mathrm{FN}}{\sqrt{\left(\mathrm{TP}+\mathrm{FP}\right)\left(\mathrm{TP}+\mathrm{FN}\right)\left(\mathrm{TN}+\mathrm{FP}\right)\left(\mathrm{TN}+\mathrm{FN}\right)}}$$

The percentage of instances correctly classified as positive is known as the true positive (TP), while the percentage of instances incorrectly classified as negative is known as the false negative (FN). The percentage of instances correctly classified as negative is known as the true negative (TN). Finally, the percentage of instances incorrectly classified as positive is known as the false positive (FP).

We also calculate feature map size with Equation (6):Feature mapsize = 1 + [input size − filter size + 2 × padding]/strid(6)

### 5.2. Quantitative Evaluation and Discussion

This study evaluates the effectiveness of the proposed CNN-LightGBM with multiple threads model using the LC25000 lung histopathology image dataset. The proposed classification scheme divided the image into two categories: benign cells and malignant cells. There are 5000 histopathological images for each class. In [7], where practical tests with various splitting percentages for training and testing datasets were performed, it was empirically demonstrated that the accuracy rates do not seem to vary considerably. Therefore, the dataset has been randomly divided into 40% and 60% of the lung images for training and testing, respectively, as shown in Table 3.

Table 4 illustrates the experimental analysis is carried out using dissimilar feature extractors in deep learning models using classification methods such as Convolutional Neural Networks (CNN), Support Vector Machines (SVM), Random Forests (RF), AdaBoost, XGBoost, and LightGBM multiple threads. The proposed CNN model, when compared to existing deep learning models, including VGG16, VGG19, AlexNet, Inception ResNet v2, ResNet50, Inception v3, GoogleNet, and MobileNet, obtained a minimum total number of training parameters of one million. In comparison to other deep learning models, it also obtained a minimum consumption time of one second during the feature extraction process. Additionally, it only obtained two seconds to classify data using LightGBM’s multiple threads.

As shown in Figure 6, the chart of time consumption for feature extraction and classification models is shown. The proposed CNN-LightGBM multiple threads model has been compared with other deep learning models, such as VGG16, VGG19, AlexNet, Inception ResNet_v2, ResNet50, Inception_v3, GoogleNet, and MobileNet for feature extraction, and with some machine learning models, such as SVM, RF, AdaBoost, XGBoost, and LightGBM with multiple threads for classification. The proposed model achieved the lowest time consumption during feature extraction and classification compared with other state-of-the-art models.

For the proposed CNN model in feature extraction and several machine learning models, we used evaluation metrics to measure efficiency. The evaluation metrics for the proposed CNN feature extractor and some existing machine learning classifiers, including SVM, RF, AdaBoost, XGBoost, and LightGBM with multi-threading, are illustrated in Table 5. The proposed CNN-LightGBM multiple threads approach has obtained 99.6, 99.6, 99.6, 99.5, and 99.1% for accuracy, F1 score, sensitivity, specificity, and MCC, respectively.

The confusion matrix for each strategy using the identical dataset is shown in Figure 7. It displays the true label versus the predicted label of the images in the supplied labeled categories for the test data. It displays the proposed CNN feature extractor model along with the currently operational machine learning classifiers. Using the LightGBM classifier, only 27 samples out of 6000 images have been misclassified. As a result, the acquired results demonstrate that ML models may be utilized to accurately and reliably categorize histpathological images of lung cancer. In the end, the experimental results from this research demonstrate that machine learning models, especially the LightGBM model, are quite good at categorizing various lung cancer subtypes, while there is still space for improvement.

## 6. Conclusions and Future Work

Worldwide, lung cancer has a very high death rate. As a result, deep learning techniques are employed to lighten the strain on pathologists and speed up the crucial process of identifying lung cancer. This research describes a unique CNN-LightGBM technique for feature extraction and classification of lung cancer histopathology images. The CNN and the LightGBM models have been combined as an innovative idea for this paper. This will affect the computation time reduction. It has been applied to the LC25000 Lung Histopathological Imaging Dataset. By examining the related pathological images, the proposed approach divides histopathological lung images into benign and malignant categories. There are 5000 histopathological images for each class of lung.

The proposed CNN-LightGBM strategy is more effective than cutting-edge methods for feature extraction and classification of histopathological lung cancer datasets. It achieved 99.6% in both accuracy and sensitivity. The lowest training parameter of one million was acquired by the proposed CNN model. Of all machine learning models, LightGBM had the fastest computation time, with only one second for training and testing datasets. This shows that, as anticipated, comparing the LightGBM tree model to the other ensemble learning methods reveals that it is generally simpler. The success of the LightGBM-based technique can be explained by the fact that light gradient boosting classifiers are a mixture of classifiers that can profit from the complementing nature of different classifiers to boost efficiency.

The findings indicate that LightGBM is an important tool for classifying medical data, particularly for the detection of patients suffering from lung cancer. The proposed CNN-LightGBM only required three seconds to extract and classify features, and it was the quickest. As a result, when compared to current methods, the suggested technique is less time consuming and more efficient at accurately classifying diseases. The experiment results show that the suggested technique performs more effectively than the majority of similar cancer diagnosis techniques in terms of time consumption. Pathologists will be able to diagnose more lung cancer patients with less effort, expense, and time in the medical centers if they use this computer-based identification method. As far as we are aware, the proposed CNN-LightGBM can be used as a benchmark result for ongoing study in histopathological cancer classification.

In future research, it will be possible to create a more lightweight model by reducing the number of parameters in the proposed CNN model. Additionally, preprocessing techniques and segmentation algorithms can be employed on the images to enhance the model’s performance and increase the accuracy for histopathology images. To improve the hyperparameters, CNN-LightGBM can also be used to classify histopathological images of different types of cancer, such as breast, prostate, and throat cancers. In order to validate the model in comparison to other models, multiclass classification may be discussed in addition to binary classification. We may also compare the performance of newer approaches like CatBoost (Categorical Boosting) or NgBoost (Natural Gradient Boosting for Probabilistic Prediction) models with that of the LightGBM classifier model.

## Figures and Tables

**Figure 1 diagnostics-13-02469-f001:**
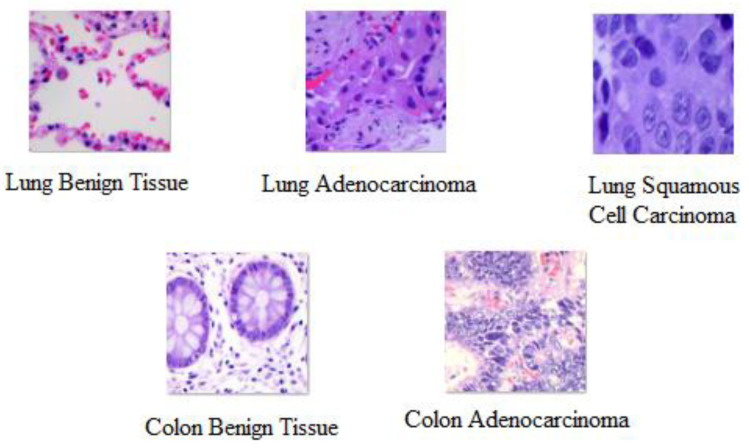
Sample images collected from the LC25000 dataset.

**Figure 2 diagnostics-13-02469-f002:**
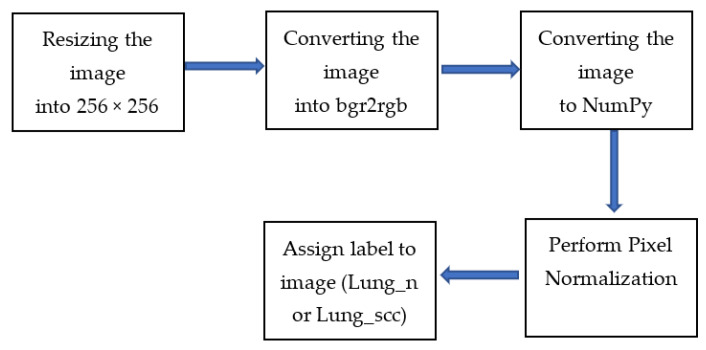
The Pre-processing Steps.

**Figure 3 diagnostics-13-02469-f003:**
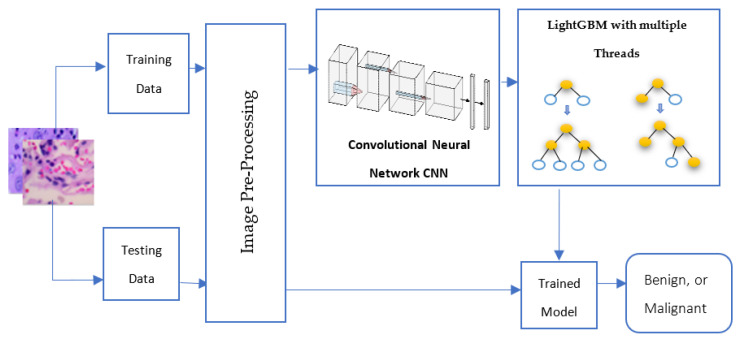
The proposed CNN-LightGBM with multiple threads model.

**Figure 4 diagnostics-13-02469-f004:**
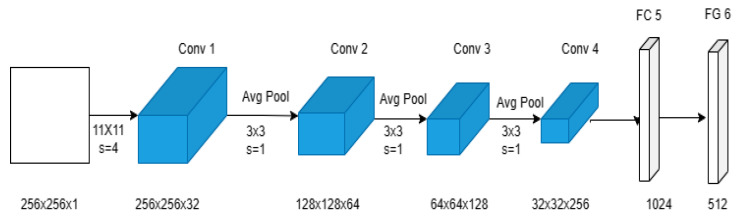
The Architecture of The Proposed CNN’s Model.

**Figure 5 diagnostics-13-02469-f005:**
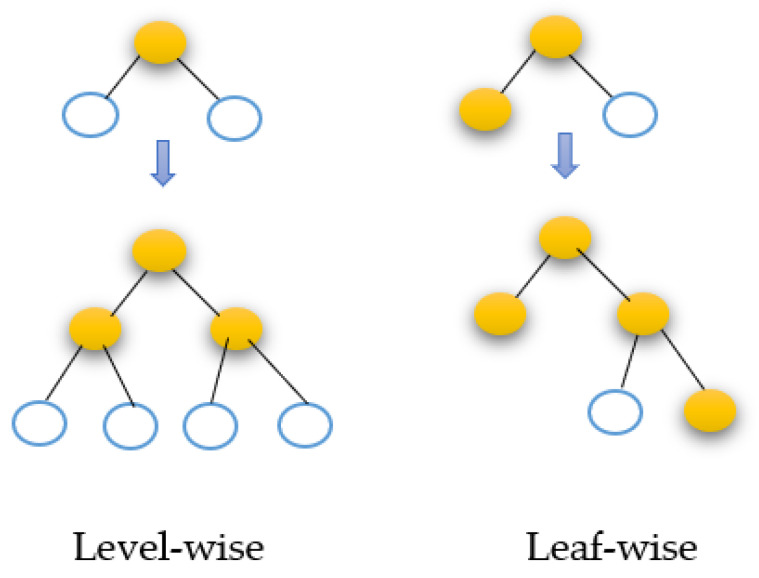
The LightGBM tree generating approach.

**Figure 6 diagnostics-13-02469-f006:**
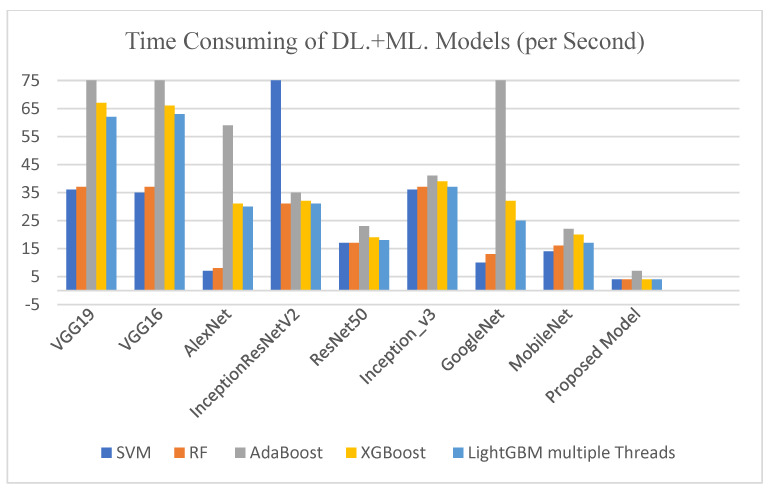
Time consumed of the proposed CNN model compared with existing DL and ML models.

**Figure 7 diagnostics-13-02469-f007:**
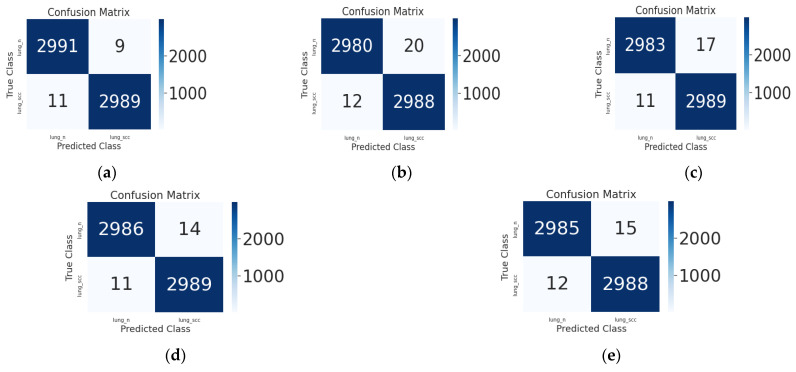
The confusion matrix display of the proposed CNN and existing ML classifiers. (**a**) SVM; (**b**) RF; (**c**) AbaBoost; (**d**) XGBoost; (**e**) LightGBM multiple Threads.

**Table 1 diagnostics-13-02469-t001:** The LC25000 dataset.

Cancer Name	Cancer Type	Label Name	Number of Samples
Colon	Adenocarcinoma	Colon_aca	5000
Colon	Benign Tissue	Colon_n	5000
Lung	Adenocarcinoma	Lung_aca	5000
Lung	Benign Tissue	Lung_n	5000
Lung	Squamous Cell Carcinoma	Lung_scc	5000

**Table 2 diagnostics-13-02469-t002:** The Set-up of the Proposed CNN Model.

Layer	#Filters/Neurons	Filter Size	Stride	#Nodes	Padding	Activation Function
Conv 1	32	11 × 11	4 × 4	-	Valid	ReLU
Max Pool1	-	2 × 2	-	-	-	-
Conv 2	64	3 × 3	1 × 1	-	-	ReLU
Max Pool2	-	2 × 2	-	-	-	-
Conv 3	128	3 × 3	1 × 1	-	-	ReLU
Max Pool3	-	2 × 2	-	-	-	-
Conv 4	256	3 × 3	1 × 1	-	-	ReLU
Max Pool4	-	2 × 2	-	-	-	-
FC 1	-	-	-	1024	-	ReLU
FC 2	-	-	-	512	-	ReLU
Dropout	Rate = 0.4	-	-	-	-	-

**Table 3 diagnostics-13-02469-t003:** The used dataset.

The Type of Cancer	Training Dataset	Testing Dataset
Lung Benign Tissue	2000	3000
Lung Squamous Cell Carcinoma	2000	3000

**Table 4 diagnostics-13-02469-t004:** The total training parameters (millions) and time consumed (seconds) of the proposed model compared with existing deep learning and machine learning models.

Algorithm	Total Parameters	Feature Mapsize	Time of Feature Extraction	Time Consuming during Classification with ML Model				
				SVM	RF	AdaBoost	XGBoost	LightGBM Multiple Threads
VGG19	171	55	32	4	5	54	35	30
VGG16	165	55	33	2	4	55	33	30
AlexNet	58	34	5	2	3	54	26	25
InceptionResNetV2	54	115	29	635	2	6	3	2
ResNet50	24	110	16	1	1	7	3	2
Inception_v3	22	115	35	1	2	6	4	2
GoogleNet	5	55	9	1	4	76	23	16
MobileNet	3	75	7	7	9	15	13	10
Proposed Model	1	64	2	2	2	5	2	1

**Table 5 diagnostics-13-02469-t005:** Evaluation metrics of the proposed CNN model with existing ML classifiers.

Proposed CNN + ML	Accuracy	F1-Score	Sensitivity	Specificity	MCC
SVM	99.6	99.6	99.6	99.7	99.3
RF	99.5	99.5	99.6	99.3	98.9
AbaBoost	99.5	99.5	99.6	99.4	99.1
XGBoost	99.6	99.6	99.6	99.5	99.1
LightGBM multiple Threads	99.6	99.6	99.6	99.5	99.1

## Data Availability

The data utilized in this work can be found at https://www.kaggle.com/datasets/andrewmvd/lung-and-colon-cancer-histopathological-images (accessed on 5 October 2022).

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
