# Peer review of "An Efficient Combination of Convolutional Neural Network and LightGBM Algorithm for Lung Cancer Histopathology Classification"

_diagnostics, 2023, doi:10.3390/diagnostics13152469_

Round 1
Reviewer 1 Report
This manuscript proposed a novel deep learning method for lung cancer histopathology classification, where CNN and LightGBM were employed for the task of interest. First, a CNN model was developed for feature extraction after preprocessing of image. Then, the extracted features were sent to LightGBM for lung tissue classification. The performance of the proposed method has been validated using experimental dataset, with satisfactory results. Overall, the topic of this research is interesting, and the manuscript was well organised and written. The detailed comments are summarised as follows.
1. The main innovation and contribution of this research should be clearly clarified in abstract and introduction.
2. Please broaden and update literature review on CNN or deep learning to demonstrate its excellent capacity to resolve practical problems. E.g. Torsional capacity evaluation of RC beams using an improved bird swarm algorithm optimised 2D convolutional neural network; Automated damage diagnosis of concrete jack arch beam using optimized deep stacked autoencoders and multi-sensor fusion.
3. The architecture of CNN model is suggested to be presented in the form of graph.
4. The performance of trained CNN model is heavily dependent on the setting of hyperparameters. How did the authors tune/optimise the network parameters to achieve the best prediction accuracy in this research?
5. This study used different pre-trained deep CNNs for feature extraction, such as AlexNet, VGG, etc. Accordingly, transfer learning should be used in this study. Please add more details on how these pretrained CNNs were transferred to the model in this research.
6. A comparison with other models from literature is necessary to prove the superiority of proposed method.
7. More future research should be included in conclusion part.
English language is good
Author Response
Thanks for your comments, please see the attachment.
Point 1: The main innovation and contribution of this research should be clearly clarified in abstract and introduction.
Response 1: Thanks for your comment, Our modification in Abstract is: In this paper, we present an innovative method for rapidly identifying and classifying histopathology images of lung tissues by combining a newly proposed CNN model with few total parameters and the enhanced Light Gradient Boosting Model (LightGBM) classifier. Our modification in Introduction is:
The primary contributions made by this study are the pre-processing LC25000 approaches used to improve the contrast between the images after they have been extracted from the lung histopathology image dataset. The efficient proposed CNN model with the lowest training parameter for obtaining discriminative feature vectors is then used to do the feature extraction. The LightGBM with the fastest computation time was utilized to classify lung tissue to increase the efficiency and accuracy of disease classification. This study's innovation is a combination of the proposed CNN-LightGBM strategy that can fast recognize and categories histological images of lung tissues. It also minimizes the false negative rate, which lowers the risk of incorrectly concluding that the disease of a patient doesn't exist. Then, we compare the proposed model with existing machine learning and deep learning techniques. The proposed CNN-LightGBM strategy is more effective than cutting-edge methods for feature extraction and classification of histopathological lung cancer datasets.
Point 2: Please broaden and update literature review on CNN or deep learning to demonstrate its excellent capacity to resolve practical problems. E.g. Torsional capacity evaluation of RC beams using an improved bird swarm algorithm optimized 2D convolutional neural network; Automated damage diagnosis of concrete jack arch beam using optimized deep stacked autoencoders and multi-sensor fusion.
Response 2: We add in Related work: The use of machine learning (ML) and deep learning (DL) to categorize and identify has been a hot topic for a while. In recent years, artificial intelligence (AI) technology based on convolutional neural network (CNN) has been widely applied in various fields [21,22]. [21] Yu, Y.; Liang, S.; Samali, B.; Nguyen, T.N.; Zhai, C.; Li, J.; Xie, X. Torsional capacity evaluation of RC beams using an improved bird swarm algorithm optimised 2D convolutional neural network. Eng. Struct. 2022, 273, 115066.
[22] Yu, Y.; Li, J.; Li, J.; Xia, Y.; Ding, Z.; Samali, B. Automated damage diagnosis of concrete jack arch beam using optimized deep stacked autoencoders and multi-sensor fusion. Dev. Built Environ. 2023, 14, 100128.
Point 3: The architecture of CNN model is suggested to be presented in the form of graph.
Response 3: Thanks for your comment, I draw the CNN architecture
Point 4: The performance of trained CNN model is heavily dependent on the setting of hyperparameters. How did the authors tune/optimize the network parameters to achieve the best prediction accuracy in this research?
Response 4: Hyperparameters are the factors that affect how the network is trained as well as the variables that govern the network's topology. Before training (and before maximizing the weights and bias), hyperparameters are established. The suggested model, which only uses four convolutional layers, four maximum pooling layers, and one leaky layer, gives us the greatest accuracy with the fewest total parameters out of all the models we tried to build. It can be argued that the proposed CNN-LightGBM with multiple threads model performs better than the majority of the current models. With the quickest computation time of only three seconds and the fewest number of parameters needed to identify and classify lung tissue, it was able to reach an accuracy of up to 99.6%.
Point 5: This study used different pre-trained deep CNNs for feature extraction, such as AlexNet, VGG, etc. Accordingly, transfer learning should be used in this study. Please add more details on how these pretrained CNNs were transferred to the model in this research.
Response 5: Thank you for your comment, however, we thoroughly investigated these pre-trained CNNs and built our own CNN model using less parameters to speed up processing time.
Point 6: A comparison with other models from literature is necessary to prove the superiority of proposed method.
Response 6: We add in Related work: Hyperparameters are the factors that affect how the network is trained as well as the variables that govern the network's topology. Before training (and before maximizing the weights and bias), hyperparameters are established. The suggested model, which only uses four convolutional layers, four maximum pooling layers, and one leaky layer, gives us the greatest accuracy with the fewest total parameters out of all the models we tried to build. It can be argued that the proposed CNN-LightGBM with multiple threads model performs better than the majority of the current models. With the quickest computation time of only three seconds and the fewest number of parameters needed to identify and classify lung tissue, it was able to reach an accuracy of up to 99.6%.
Point 7: More future research should be included in conclusion part. Response 7: Thanks for your comment, we add in the conclusion:
In the future research, it is possible to create a more lightweight model by reducing the number of parameters in the proposed CNN model. Additionally, preprocessing techniques and segmentation algorithms can be employed on the images to enhance the model's performance and increase the accuracy for histopathology images. To improve the hyperparameters, CNN-LightGBM can also be used to classify histopathological images of different types of cancer such as breast, prostate, and throat cancers. In order to validate the model in comparison to other models, multiclass classification may be discussed in addition to binary classification. We may also compare the performance of newer approaches like CatBoost (Categorical Boosting) or NgBoost (Natural Gradient Boosting for Probabilistic Prediction) models with that of the LightGBM classifier model.

Reviewer 2 Report
This manuscript proposed a method to classify lung cancer using histopathology images both efficiently and accurately. The authors combined deep learning and machine learning methods to build a new architecture for histopathology classification. This manuscript has certain research values. However, for further improvement, here are some suggestions :
1. In line 69-73, when introducing CNN in medical tasks, it is recommended to add some introduction for latest segmentation research, for example: (IEEE Transactions on Medical Imaging, 2022, Generative consistency for semi-supervised cerebrovascular segmentation from TOF-MRA; ACM ICMR 2022, 2022, pp.668-676, Weakly-supervised cerebrovascular segmentation network with shape prior and model indicator; Nature methods, 18(2), 2021, pp.203-211, nnU-Net: a self-configuring method for deep learning based biomedical image segmentation)
2. There are many repetitive abbreviations in the text, for example, in line 73 and line 77, the full name of CNN appears twice; in line 164 and line 173, the full name of SCC appears twice, … etc., it would be more academic if the authors give the full name just one time.
3. In line 104, the authors said that ‘Inception-ResNetv2 worked exceptionally well in this research, with the accuracy of 99.7%’, what does ‘this research’ refer to? If it refers to this specific manuscript, the authors seem to forget to put this result in Table 5.
4. In line 134, I strongly suggest the authors to add the full name of EGOA-random forest here, since it is the first time the abbreviation ‘EGOA’ shows up.
5. In line 159, the authors said ‘employed for augmentation’, I believe it would be better if the authors can explain what exact augmentations they have employ.
6. The Figure 1 is too blurry, I recommend the authors to replace it with a more HD version.
7. In Figure 2, the authors call the 4th pre-processing step ‘Feature Scaling’, from the text explanation, I do believe if the authors could choose the name ‘Pixel Normalization’ would make this manuscript more academic.
8. The authors choose LightGBM as their ML method, as far as I know, LightGBM is a relatively old method, why not use newer methods like CatBoost or NgBoost? If the author can explain this would be better.
9. In line 240, I think the sentence ‘Layers and parameters. are …’ may occur a wrong use of punctuation.
10. In Section 5, the authors designed a proper experiment to show strength of their proposed method. However, they have not evaluated their method with a k-fold manner, I believe this manuscript would be more valuable if they can do this.
11. In Table 5, the name of ‘XGBoost’ is wrongly spelled as ‘XGBBoost’. Also, the metrics of XGBoost and LightGBM are exactly the same, I wonder if it is a clerical error or the experimental results?
Minor editing of English language required.
Author Response
Thanks for your comments, please see the attachment.

Reviewer 3 Report
Paper presents binary classification approach for lung cancer using its histopathological images. Features are obtained using CNN and LightGBM is used as a classifier. A comparison of learnable parameters, and execution time with tradition CNNs and traditional machine learning models are presented in the paper. Paper is written well but most important part of the paper i.e. comparison with literatures is missing. Many literatures claims similar accuracy. So it is required to explain why this model is important then other models. I suggest to refer paper https://doi.org/10.3390/diagnostics13091594 .
A discussion on multiclass classification can be provided instead of binary to validate model in compared to other models.
A comparison with standard CNN network is not appropriate. I suggest to analyze results reducing layers of VGG, ResNET etc and a discussion can be added to show appropriate comparison.
Express GPU version used because timings depend on machine used.
The comparison shown in Table 4 is confusing. The classification time depends on size of features sets. It is more appropriate to presents features size in the table. What Column 4-8 represents?
Author Response
Thanks for your comments, please see the attachment.
Point 1: A comparison of learnable parameters, and execution time with tradition CNNs and traditional machine learning models are presented in the paper. Paper is written well but most important part of the paper i.e., comparison with literatures is missing. Many literatures claim similar accuracy. So, it is required to explain why this model is important than other models.
Response: Thanks for your comment, Our modification in the background: Hyperparameters are the factors that affect how the network is trained as well as the variables that govern the network's topology. Before training (and before maximizing the weights and bias), hyperparameters are established. The suggested model, which only uses four convolutional layers, four maximum pooling layers, and one leaky layer, gives us the greatest accuracy with the fewest total parameters out of all the models we tried to build. It can be argued that the proposed CNN-LightGBM with multiple threads model performs better than the majority of the current models. With the quickest computation time of only three seconds and the fewest number of parameters needed to identify and classify lung tissue, it was able to reach an accuracy of up to 99.6%.
Point 2: I suggest to refer paper https://doi.org/10.3390/diagnostics13091594.
Response: We add in the Related work: EffcientNetV2 big, medium, and small models are a deep learning architecture built on the concepts of compound scaling and progressive learning [33]. Using the EffcientNetV2-L model for the 5-class categorization of lung and colon cancers, they attained an accuracy of 99.97% on the test set. [33] TUMMALA, Sudhakar, et al. An Explainable Classification Method Based on Complex Scaling in Histopathology Images for Lung and Colon Cancer. Diagnostics, 2023, 13.9: 1594.
Point 3: A discussion on multiclass classification can be provided instead of binary to validate model in compared to other models.
Response: Thanks for your comment, due to the current time constraints, we will take your comment in consideration in the future research, in order to validate the model in comparison to other models, multiclass classification may be discussed in addition to binary classification.
Point 4: A comparison with standard CNN network is not appropriate. I suggest to analyze results reducing layers of VGG, ResNET etc and a discussion can be added to show appropriate comparison.
Response: Thanks for your comment, however, this is our contribution as we propose CNN model with a few numbers of total parameters and achieve high accuracy with the quickest computation time.
Point 5: Express GPU version used because timings depend on machine used.
Response: GPU version used is NVIDIA T4 Tensor Core GPUs.
Point 6: The comparison shown in Table 4 is confusing. The classification time depends on size of features sets. It is more appropriate to presents features size in the table. What Column 4-8 represents?
Response: Thanks for your comment, we calculated feature size and added it in table: feature size = 1+ [input size-filter size +2*padding]/strid Column 4-8 represents the time consuming in seconds during classification with ML model

Round 2
Reviewer 1 Report
The authors have well addressed the reviewer's comments. Hence, I suggest that this revised version can be accepted for publication.
Author Response
Thank you for your suggestion.
Reviewer 2 Report
Most of my concerns have been completed. There are only small suggestions:
1. In line 87, the full name of CNN appears. However, the abbreviated name CNN was already used in line 69, line 70 and line 72. Please make sure the abbreviation is not appeared before its full name. Also, in line 114 the full name of CNN shows again, the authors may want to fix it.
2. In line 112, the full name of deep learning shows up the first time, but in line 258 it shows again. It seems this manuscript has too many repetitive full names, it would be better if the authors can check it carefully.
Minor editing of English language required.
